# Fingolimod Inhibits Exopolysaccharide Production and Regulates Relevant Genes to Eliminate the Biofilm of *K. pneumoniae*

**DOI:** 10.3390/ijms25031397

**Published:** 2024-01-23

**Authors:** Xiang Geng, Ya-Jun Yang, Zhun Li, Wen-Bo Ge, Xiao Xu, Xi-Wang Liu, Jian-Yong Li

**Affiliations:** Key Laboratory of New Animal Drug Project of Gansu Province, Key Laboratory of Veterinary Pharmaceutical Development of Ministry of Agriculture and Rural Affairs, Lanzhou Institute of Husbandry and Pharmaceutical Sciences of Chinese Academy of Agricultural Sciences, Lanzhou 730050, China; gengxiang180@163.com (X.G.); yanyue10224@163.com (Y.-J.Y.); zhun2828@163.com (Z.L.); gewenbo@caas.cn (W.-B.G.); xu05xiao@163.com (X.X.)

**Keywords:** *K. pneumoniae*, biofilm, fingolimod (FLD), antibacterial, synergistic

## Abstract

*Klebsiella pneumoniae* (*K. pneumoniae*) exhibits the ability to form biofilms as a means of adapting to its adverse surroundings. *K. pneumoniae* in this biofilm state demonstrates remarkable resistance, evades immune system attacks, and poses challenges for complete eradication, thereby complicating clinical anti-infection efforts. Moreover, the precise mechanisms governing biofilm formation and disruption remain elusive. Recent studies have discovered that fingolimod (FLD) exhibits biofilm properties against Gram-positive bacteria. Therefore, the antibiofilm properties of FLD were evaluated against multidrug-resistant (MDR) *K. pneumoniae* in this study. The antibiofilm activity of FLD against *K. pneumoniae* was assessed utilizing the Alamar Blue assay along with confocal laser scanning microscopy (CLSM), scanning electron microscopy (SEM), and crystal violet (CV) staining. The results showed that FLD effectively reduced biofilm formation, exopolysaccharide (EPS), motility, and bacterial abundance within *K. pneumoniae* biofilms without impeding its growth and metabolic activity. Furthermore, the inhibitory impact of FLD on the production of autoinducer-2 (AI-2) signaling molecules was identified, thereby demonstrating its notable anti-quorum sensing (QS) properties. The results of qRT-PCR analysis demonstrated that FLD significantly decreased the expression of genes associated with the efflux pump gene (*AcrB*, *kexD*, *ketM*, *kdeA*, and *kpnE*), outer membrane (OM) porin proteins (*OmpK35*, *OmpK36*), the quorum-sensing (QS) system (*luxS*), lipopolysaccharide (LPS) production (*wzm*), and EPS production (*pgaA*). Simultaneously, FLD exhibited evident antibacterial synergism, leading to an increased survival rate of *G. mellonella* infected with MDR *K. pneumoniae*. These findings suggested that FLD has substantial antibiofilm properties and synergistic antibacterial potential for colistin in treating *K. pneumoniae* infections.

## 1. Introduction

*K. pneumoniae* is a frequently encountered pathogen in clinical settings and a significant cause of hospital-acquired infections [1]. *K. pneumoniae* typically does not cause disease in healthy individuals. However, it can evoke primary pneumonia, meningitis, urinary tract infections, and traumatic infections, particularly in individuals with compromised immunity [2,3,4]. The prevalence of *K. pneumoniae* and its drug resistance is steadily increasing. The overuse of antibiotics and bacterial mutation have contributed to the emergence of multidrug-resistant *K. pneumoniae*, which is one of the most significant global issues today [5,6]. According to the National Institute of Health (NIH), bacterial infections cause over 16 million deaths globally annually [7]. Recent research has provided evidence of a strong correlation between antimicrobial resistance in bacteria and biofilm formation, which is responsible for approximately 75% of human bacterial infections [8,9].

*K. pneumoniae* exhibits a propensity for biofilm formation and the layer-by-layer encapsulation of bacteria [10]. The extracellular polymeric matrix that reinforces complex biofilm consists of proteins, lipids, carbohydrates, and DNA, all of which contribute to heightened resistance against antibiotics [11]. Bacterial cells within biofilms exhibit reduced sensitivity to antibiotics compared to their planktonic counterparts [12], thereby enhancing resistance to antimicrobial agents and impeding the efficacy of treatments for biofilm-related infections. Numerous studies have demonstrated a resistance capacity of biofilm bacteria that is over 1000 times greater than that of planktonic bacteria when exposed to antibiotics [13,14]. Biofilms consist of complex assemblages of microorganisms and exopolymeric materials that undergo stages of accumulation, attachment, maturation, and dispersal [15]. The process of biofilm formation is influenced by cell and matrix characteristics, environmental factors, and genetic regulation within bacterial populations [16]. Biofilms enable bacterial adhesion to substrates and their self-encapsulation within a matrix composed of EPS, proteins, and extracellular DNA (eDNA) [12,17]. EPS, serving as the foremost structural component of biofilm, generates a protective shield on the surface of microorganisms [18]. This shield provides protection against adverse environmental conditions that facilitate the dissemination of foodborne disease, simultaneously bolstering bacterial and innate immune responses [19,20,21]. Proteins, polysaccharides, and eDNA represent the primary structural constituents of EPS [22]. Polysaccharide intercellular adhesin (PIA), a linear glucosamine polymer, constitutes the predominant element of polysaccharide slime, which facilitates bacterial cell adhesion and the accumulation of biofilm [23]. Extracellular DNA plays a pivotal role in bacterial attachment during both the formation and maturation stages of biofilm development [24]. Research conducted domestically and internationally has shown that inhibiting the synthesis of polysaccharide proteins by bacteria can effectively inhibit the formation of biofilms [25]. Biofilms consist of high concentrations of cells that regulate gene expression depending on cell density via quorum sensing (QS). According to the US National Institutes of Health, approximately 80% of microbial infections are attributed to biofilms that are regulated by QS molecules, which enable cells to detect population density and modulate the expression of genes associated with target functions by interacting with transcription factors [26,27,28]. Therefore, the inhibition and elimination of biofilms are highly relevant to the successful treatment of bacterial infections.

Recently, these have been numerous reports in the literature on the use of compounds from various sources for antibiofilm purposes. Antimicrobial peptides such as protegrin 1, pleurocidin, ll-37, and human β defensin 3 [29], along with nanoparticles such as ZIF-67 nanoparticles consisting of imidazolate anions and cobalt cations [30] and nanostructured copper films [31], have been investigated for their potential antibiofilm properties. Additionally, natural products and their derivatives, including carvacrol, cinnamaldehyde, thymol [32], and fingolimod (FLD) [33], have shown promising antibiofilm activity. FLD is an FDA-approved oral drug for multiple sclerosis that regulates sphingosine-1-phosphate (S1P) receptors to reduce lymphocyte circulation [34,35]. In a recent study, it was discovered that FLD exhibited substantial antibacterial activity against Gram-positive bacteria, including *Staphylococcus aureus* (*S. aureus*) and *Acinetobacter baumannii* (*A. baumannii*), and demonstrated specific antibiofilm effects on Gram-positive bacteria. Previous studies have demonstrated the synergistic antimicrobial effects of FLD; however, its activity against *K. pneumoniae* biofilm has not yet been reported [33]. Alternatively, screening active molecules from approved drugs offers two obvious advantages: established safety and time-saving efficiency.

This study aimed to assess the antibiofilm properties of FLD against *K. pneumoniae*, investigate its mechanisms of action, and evaluate its potential as an adjunct therapy for drug-resistant strains. qRT-PCR was employed to analyze the altered gene expression of efflux pump-associated genes (*AcrA*, *AcrB*, *kexD*, *ketM*, *kdeA*, and *kpnE*), OM porin protein genes (*OmpK35*, *OmpK36*), lipopolysaccharide production (*wzm*), QS system (*luxS*), EPS production (*pgaA*), and type III fimbriae (*mrkA*) in response to FLD treatment for inhibiting *K. pneumoniae* biofilm formation.

## 2. Results

### 2.1. FLD Had no Effect on Growth and Metabolic Activity

The effect of FLD on the growth and metabolic activity of *K. pneumoniae* was measured by micro broth dilution and the Alamar Blue (AB) assay. The results showed that FLD had no significant effect on the growth of *K. pneumoniae* in the concentration range of 4 to 64 µg/mL compared to the control group. However, the bacterial metabolic activity was significantly decreased when the concentration was greater than 32 µg/mL FLD (Figure 1A). These results revealed that the non-bactericidal nature and unaltered metabolic activity of *K. pneumoniae* were not markedly affected by FLD concentrations no greater than 32 µg/mL.

### 2.2. FLD Can Inhibit Biofilm Formation and Eradicate Mature Biofilm

Different concentrations of FLD (0.25, 0.5, 1, 2, 4, 8, 16, and 32 µg/mL) were measured for antibiofilm activity using the CV staining method. This study showed that FLD treatment significantly inhibited the formation of *K. pneumoniae* in a dose-dependent manner (Figure 1B). Even 1 µg/mL of FLD was also found to be effective against the formation of biofilm (inhibition rate ≥ 20%). The inhibition rate of biofilm formation was 21.85%, 29.77%, 43.94%, and 45.72% at 2, 4, 8, 16, and 32 µg/mL, respectively.

For the existing mature biofilm of *K. pneumoniae*, the eradication activity of FLD was also evaluated. The results showed that as the concentration of FLD increased, the eradication of biofilms also increased, suggesting a dose-dependent relationship between FLD concentration and eradication of biofilms (Figure 1C). FLD could significantly eradicate mature biofilm even at 0.5 μg/mL, with an eradication rate of over 20%. Furthermore, FLD was able to eradicate over 80% of the biofilms at 128 μg/mL.

### 2.3. CLSM Analysis

To further observe the biofilm-inhibitory effect of FLD, CLSM imaging analysis was conducted using the fluorescent dye SYTO-9 and propidium iodide (PI). The result showed that FLD significantly inhibited biofilm formation, as demonstrated by a negative gradient in green fluorescence in *K. pneumoniae* strains (Figure 2). The results showed that FLD inhibited biofilm formation. The results revealed a tight biofilm structure, bright green fluorescence, and bacterial aggregation, indicating well-developed biofilms in the control group. However, when the concentration of FLD increased, the fluorescent properties shifted from green to red. These results suggest that FLD disrupts biofilm integrity and inhibits biofilm formation, which has a detrimental effect on biofilm formation.

### 2.4. SEM Analysis

The inhibition of biofilm formation by FLD and its relationship to bacterial morphology was investigated. SEM results showed that the control group exhibited a predominantly uniform distribution of elliptical or rod-shaped cells, with little cell accumulation. However, after FLD treatment, significant changes were observed in comparison to the control group. Specifically, the application of FLD at a concentration of 32 μg/mL had a significant impact on cell morphology. In the 10 µm image, it is evident that after FLD treatment, cells tended to aggregate and form loosely arranged and stacked micro-clusters, creating a three-dimensional mesh-work. Additionally, some cell membrane structures appear to be disrupted, leading to cell death. Similarly, in the 5 µm and 2 µm images, the cell membrane exhibits significant shrinkage and an increase in cell aggregation (Figure 3).

### 2.5. FLD Can Decrease the Production of EPS

As the most important structure of biofilms, EPS can protect cells and enhance bacterial tolerance and innate immune response [36]. To access the production of EPS in biofilms after treatment with FLD, the ruthenium red staining method was used. The results demonstrated that FLD treatment effectively inhibited EPS production in a dose-dependent manner (Figure 4A). Even at a low concentration of 4 µg/mL, FLD was found to significantly inhibit EPS production, with an inhibition rate exceeding 20%. The results obtained are consistent with the inhibitory effect of FLD on biofilm formation. The results showed that FLD reduced the EPS production of *K. pneumonia*.

### 2.6. Effects of FLD Can Inhibit the Motility of K. pneumonia

There is a close relationship between the bacterial biofilm formation and motility of *K. pneumonia*. Thus, the effect of FLD on motility was evaluated according to previous studies [37]. As showcased in Figure 4B, the size of the halo zone was measured quantitatively, and it was observed that FLD treatment with different concentrations resulted in a significantly smaller halo zone (*p* < 0.01). The reduction in the size of the halo zone with increasing FLD concentrations suggests a dose-dependent inhibitory effect on biofilm formation. This indicates that FLD effectively inhibited the growth and expansion of biofilms formed by *K. pneumoniae*.

### 2.7. FLD Can Reduce the Count of Viable Cells in Biofilms

Bacteria in biofilms were enumerated by quantitative plate count [38]. After incubation at 37 °C for 24 h, the number of bacterial colonies was recorded and expressed in log_10_ CFU/mL. As shown in Figure 4C, all FLD treatments significantly decreased the number of bacteria in the biofilm models (*p* < 0.0001). The number of bacterial cells decreased with increasing FLD concentration.

### 2.8. Effect of FLD on AI-2 Production

The formation of biofilms by bacteria is contingent upon QS [39]. To evaluate the effect of FLD on AI-2 production induced by *K. pneumoniae*, a bioluminescence assay of *V. harveyi* BB170 was employed. As shown in Figure 4D, FLD treatment led to a 15.4% inhibition in AI-2 production when administered at a concentration of 8 µg/mL in comparison to the control group. Additionally, the inhibitory effect of FLD on bioluminescence exhibited a dose-dependent relationship, with an IC50 value estimated to be 140.6 µg/mL. This finding suggests that FLD has the potential to disrupt QS, a crucial mechanism for biofilm formation and virulence factor expression in *K. pneumoniae*.

### 2.9. FLD Diminishes the Content of Biofilm Components

#### 2.9.1. Total Lipids

After being treated with different concentrations of FLD, there was an inhibitory effect on the formation of total lipids. As shown in Figure 5A, the amount of total lipids decreased with increasing FLD concentration. The inhibitory effect of FLD on total lipids was concentration-dependent. Specifically, when treated with FLD at 4 μg/mL, the inhibitory effect on total lipids exceeded 10%, and this difference was statistically significant compared to the control group (*p* < 0.0001). These findings highlight the ability of FLD to interfere with lipid metabolism in *K. pneumoniae*.

#### 2.9.2. Polysaccharide Slime

The treatment with different concentrations of FLD resulted in a noticeable inhibitory effect on polysaccharide slime formation, as shown in Figure 5B. Specifically, biofilm cells treated with 4 µg/mL FLD exhibited a significant reduction in polysaccharide slime compared to the control group (*p* < 0.0001). These findings suggest that FLD has the potential to interfere with the production of polysaccharide slime, which is a crucial component of biofilm formation in *K. pneumoniae*.

#### 2.9.3. Total DNA and eDNA

The effects of FLD treatment on the content of both total DNA and eDNA were evaluated and are shown in Figure 5C,D. The results indicate that the amount of total DNA decreased with increasing concentrations of FLD. Notably, the reduction rate of total DNA at 4 μg/mL FLD was significantly different from the blank treatment group. Regarding eDNA production, the 4 μg/mL FLD treatment group showed some inhibition, although there was no significant difference (*p* = 0.1384). On the other hand, other concentrations of FLD treatment groups exhibited increased inhibition of eDNA with higher FLD concentrations. In summary, FLD demonstrated concentration-dependent effects on both total DNA and eDNA production. These findings suggest that FLD has the potential to interfere with DNA synthesis and release mechanisms in *K. pneumoniae* biofilms.

### 2.10. Effects of Drug Treatments on Biofilm-Associated Gene Expression

To investigate the inhibitory effects of FLD on biofilm formation, the transcription levels of various genes associated with efflux pump-associated genes (*AcrA*, *AcrB*, *kexD*, *ketM*, *kdeA*, and *kpnE*), OM porin protein genes (*OmpK35*, *OmpK36*), an LPS production gene (*wzm*), a quorum-sensing (QS) system gene (*luxS*), an EPS production gene (*pgaA*), and a type III fimbriae gene (*mrkA*) were evaluated using qRT-PCR after treatment with a concentration of FLD (32 µg/mL). The results, as shown in Figure 6A, demonstrated a significant reduction in the expression of these genes in the presence of FLD compared to the control group (*p* < 0.0001), except for *AcrA* and *mrkA*. FLD effectively inhibited the formation of biofilms in *K. pneumoniae* by suppressing the transcription of key genes involved in the efflux pump, OM porin protein, adhesion, and QS system. These findings suggest that FLD has a broad inhibitory effect on various genetic pathways associated with biofilm formation in *K. pneumoniae*.

### 2.11. Safety Assay

After injecting different doses of FLD into *G. mellonella*, the insects were observed for 120 h. No deaths occurred during the experiment in the group treated with 128 mg/kg of FLD. However, two deaths were observed in the groups treated with PBS and 256 mg/kg of FLD, respectively. The survival rates of the two groups (PBS and 256 mg/kg) were the same within the 120 h observation period (Figure 6B). This suggests that FLD is safe for *G. mellonella* and could be utilized in the subsequent experiments.

### 2.12. Survival Rate of G. mellonella

The *G. mellonella* were inoculated with approximately 10^6^ CFU of bacteria and randomized to various treatment groups. Data were collected every 12 h for a total of 96 h. As expected, there were no deaths in the group treated with PBS alone throughout the 4-day experiment. In contrast, all *G. mellonella* in the model, colistin, and FLD groups died from bacterial infection within 36 h, although with distinct processes. This is consistent with the pathogenic nature of the bacterial infection in the model. However, when FLD and colistin were combined, a reduction in mortality was observed. About 20% of the *G. mellonella* died within 48 h, but no further deaths occurred thereafter (Figure 6C). The combination of FLD and colistin may have enhanced the antimicrobial activity against the bacterial infection, leading to improved survival rates in *G. mellonella*.

## 3. Discussion

Anti-virulence and antibiofilm treatments are important strategies to combat bacterial infection [40,41]. Bacterial biofilms are complex structures formed by microbial communities that are embedded in a self-produced extracellular polymer matrix. Biofilms are notorious for their resistance to antimicrobial agents, and the development of biofilms results in significant challenges in the effectiveness of antimicrobial therapy [42]. In recent years, there has been an increasing interest in the exploration of natural products with antibiofilm properties. Many studies have focused on understanding the formation of biofilm by microorganisms such as *Staphylococcus aureus*, *Pseudomonas aeruginosa*, and *Candida albicans*. However, there is comparatively little research on the development of biofilms by *K. pneumoniae*, another important pathogenic bacterium [43]. In this context, the present study investigated the effects of FLD on biofilm formation and disruption in *K. pneumoniae*. According to the reference, the *K. pneumoniae* biofilm models were established [44]. The results of the crystal violet (CV) test showed that FLD significantly inhibited the biofilm formation of *K. pneumoniae*, and this disruption was found to be dose-dependent. Furthermore, when FLD was added after biofilm formation, at a concentration of 1 µg/mL, it was observed to have a significantly disruptive effect on the mature biofilm, similar to its inhibitory concentration. The results showed that FLD significantly inhibited the formation of biofilms and disrupted mature biofilm of *K. pneumoniae* without visibly affecting bacterial growth. This suggests that FLD not only prevents the formation of biofilms but also has the ability to disrupt existing biofilms. Importantly, this study found that FLD had these inhibitory and disruptive effects on *K. pneumoniae* biofilms without visibly affecting bacterial growth. This suggests that FLD specifically targets the biofilm formation and structure of *K. pneumoniae* without impacting its overall growth. These findings further support the potential of FLD as an effective antibiofilm agent against *K. pneumoniae* infections. However, more research is needed to fully understand the underlying mechanisms of FLD’s inhibitory and disruptive effects on biofilms and to evaluate its efficacy in different models and clinical settings.

The microstructure results in conjunction with the quantitative findings provided further evidence of the inhibition and elimination activities of FLD against biofilm. The consistent results from different analysis techniques, such as CV quantification, confocal laser scanning microscopy (CLSM), and scanning electron microscopy (SEM), validate the effectiveness of FLD in inhibiting and eliminating *K. pneumoniae* biofilms. One key advantage of FLD, as mentioned, is that it acts differently from conventional antimicrobial agents. While traditional agents primarily target bacterial growth or kill bacteria, FLD specifically focuses on inhibiting and disrupting biofilm formation. This unique mechanism may decrease the risk of developing resistance in *K. pneumoniae* biofilms. Bacterial biofilms are known to contribute to chronic infections due to their increased resistance to antimicrobial agents, tolerance to treatments, and ability to evade host defense mechanisms [45]. By specifically targeting biofilm formation and structure, FLD may offer a novel approach to combat *K. pneumoniae* infections and potentially reduce the development of drug-resistant biofilms. However, further research is necessary to fully understand the mechanisms underlying FLD’s antibiofilm activity and to evaluate its efficacy and safety in clinical settings.

The presence of EPS in bacterial biofilm poses a challenge for antimicrobial agents in penetrating the biofilm depth. EPS acts as a physicochemical barrier, impeding the efficient delivery of antimicrobial agents by acting as a diffusion barrier, molecular sieve, and adsorbent [46,47]. EPS also plays several vital roles in biofilm formation, including nutrient and metabolite transport [48], maintenance of the biofilm structure [49], protection against external factors [50], and facilitating bacterial adherence and colonization on abiotic surfaces [51]. Our results demonstrate that FLD inhibits EPS production in a dose-dependent manner, thereby reducing bacterial colonization and biofilm formation. The inhibition of EPS production by FLD may disrupt the function of EPS, leading to a reduction in biofilm formation [33]. Specifically, FLD prevents the formation of adhesion molecules, which are essential for initial bacterial attachment and biofilm formation. To verify whether this inhibition translates into decreased viable cell counts (CFU/mL) in biofilm, the viable cell counts in biofilms were determined after 24 h of FLD treatment. Consistently with previous results, FLD reduced CFU counts, indicating a reduction in viable bacterial cells within the biofilm [33]. The formation of *K. pneumoniae* biofilm is a complex process. Lipids, polysaccharide slime, and DNA are integral components of biofilm formation. Some studies indicate that polysaccharide slime secreted in the matrix can provide shape and support to the biofilms and protect bacteria within it [19,52,53]. Therefore, inhibiting the synthesis of polysaccharide slime plays an important role in reducing biofilm formation. These components are integral to biofilm formation and stability. The reduction in their accumulation limits the formation and integrity of the biofilm structure. By inhibiting EPS production and reducing the accumulation of vital biofilm components, FLD effectively limits the formation and development of *K. pneumoniae* biofilms. These findings highlight the potential of FLD as an effective agent for preventing and disrupting biofilm formation in *K. pneumoniae* infections.

Significantly, *K. pneumoniae* has multiple virulence factors, including bacterial fimbriae, LPS, efflux pump, and siderophores, which protect the bacteria from host immune response and antibiotics and increase adherence to medical equipment and epithelial tissue surfaces [54]. The results of qRT-PCR revealed that efflux pump-associated genes (*AcrA*, *AcrB*, *kexD*, *ketM*, *kdeA*, and *kpnE*), OM porin protein genes (*OmpK35*, *OmpK36*), an LPS production gene (*wzm*), and a quorum-sensing system gene (QS) (*luxS*) were all significantly downregulated after treatment with FLD, except for those related to EPS production (*pgaA*) and type III fimbriae (*mrkA*). Gram-negative bacterial multidrug efflux is a defense mechanism that leads to antibiotic resistance and biofilm formation linked to the efflux pump system [44]. AcrB is the proton/drug antiport process’ drug specificity and energy transduction center. FLD inhibits the expression of *AcrB*, which could lead to severe impairment in resistance [55]. Similarly, the expression levels of *kexD*, *ketM*, *kdeA*, and *kpnE* indicate that RND, MATE, and SMR efflux pumps could be inhibited, which results in lower drug resistance when treated by FLD. However, FLD increased *AcrA* upregulation; this may be because bacteria tend to induce the upregulated expression of *AcrA*, consequently enhancing biofilm formation, to fight against the antibiofilm effect of FLD. Although it has been shown that the high expression of the AcrAB efflux pump confers resistance [56], this finding suggests that elevated resistance to antibiotics was not associated with an increased expression of AcrA. The AcrA pump appears to have a different role than exporting antimicrobial agents.

Multidrug-resistant (MDR) strains primarily use *mrkA* and/or *pgaA* to adhere and form biofilm; the type III fimbriae *mrkA* is associated with the formation of biofilms [44]. *K. pneumoniae* strains expressing *mrkA*, *mrkD*, and *rmpA* might be able to show moderate-to-strong biofilm formation capabilities [57]. However, FLD had no significant effect on *mrkA*, indicating that *K. pneumoniae* strains still have some ability to form biofilm. In other words, if there is no adhesion during this step, the biofilms cannot continue to form. Poly-b-1,6-N-acetyl-D-glucosamine (PGA) is a key polysaccharide in biofilm formation that involves irreversible cell–cell adhesion and attachment, stabilizing biofilm formation [58]. FLD suppresses the expression of the *pgaA* gene, inhibiting biofilm formation and corresponding with EPS production results. Interestingly, FLD downregulates the genes of the OM porin proteins *OmpK35* and *OmpK36*, suggesting that FLD inhibits OM porin proteins and suppresses *K. pneumoniae* strains’ ability to generate denser and more complex biofilms.

LPS is the primary component of the Gram-negative bacteria that acts as an endotoxin and causes fever in humans [59]. The relationship between LPS and biofilm formation is noteworthy [60]. Furthermore, the QS signal system appears to contribute to biofilm formation, with *luxS* transcripts being expressed more with bacterial growth. QS can be involved in biofilm formation through LPS biosynthesis. *LuxS* and AI-2 transport system mutation increased the expression of two LPS-synthesis genes: *wbbM* and *wzm* [61]. Consistent with this observation, FLD downregulates *luxS* and *wzm* genes while suppressing AI-2 production. FLD can alter the transcription of key genes that are regulated by affecting the efflux pump, LPS biosynthesis, and QS system.

To further evaluate the safety of FLD, in vivo experiments were conducted using *G. mellonella*. The *G. mellonella* were treated with different concentrations of FLD and monitored for any lethal effects. It was found that FLD at concentrations of 256 mg/kg had no toxic effects on the larvae, indicating that FLD was non-toxic within the effective concentration range. Moreover, this study also investigated the synergistic effect of FLD in combination with colistin in vivo. The results showed a significant increase in the survival rate when FLD was combined with colistin, indicating a synergistic effect in combating *K. pneumoniae* infections. Overall, the antibiofilm property of FLD alone was remarkable against *K. pneumoniae* in vitro and a synergistic antibacterial effect of FLD combined with colistin was found in vivo. Additionally, this study highlighted the potential of FLD as a safe and effective treatment option for combating *K. pneumoniae* infections. However, further research is needed to fully understand the mechanisms underlying the antibiofilm properties of FLD and to evaluate its efficacy in other models and clinical settings.

Our study shows that FLD successfully inhibits and destroys *K. pneumoniae* biofilm formation without affecting bacterial growth. FLD significantly decreases EPS, lipids, polysaccharide slime, DNA, and viable bacteria quantities in biofilm. According to the qRT-PCR results, FLD alters transcriptional key genes regulating biofilm formation by affecting the efflux pump (*AcrB*, *kexD*, *ketM*, *kdeA*, and *kpnE*), LPS biosynthesis (*wzm*), and QS system (*luxS*). These findings suggested that FLD had substantial antibiofilm properties in vitro, and an antibacterial synergistic potential in treating *K. pneumoniae* in vivo when combined with colistin. The combination can enhance therapeutic effects by reducing biofilm-associated drug resistance.

## 4. Materials and Methods

### 4.1. Reagents and Bacterial Strains

Clinical strains of *K. pneumoniae* were kindly provided by Professor Du Xiangdang of Henan Agricultural University (Zhengzhou, China), and all strains were cultured under conditions in MHB medium (LA, HuanKai Microbial, Guangdong, China). *Vibrio harveyi* BB170 (*V. harveyi* BB170) and *V. harveyi* BB152 were gifts from Researcher Han Xiangan (Shanghai Veterinary Research Institute, Shanghai, China, Chinese Academy of Agricultural Sciences, Beijing, China).

FLD and ruthenium red were purchased from Shanghai Yuanye Bio-technology Co., Ltd. (Shanghai, China), and it was dissolved in distilled water. A LIVE/DEAD BacLight bacterial viability kit was purchased from Invitrogen. TIANcombi DNA Lyse & Det PCR Kit was purchased from Tiangen Biotech. *G. mellonella* were purchased from Tianjin Huiyude Biotech Company. Alamar Blue (AB) was purchased from Thermo Fisher Scientific (Shanghai, China), and 0.1% crystal violet was purchased from Beijing Solaibao Biological Technology Co., Ltd. (Beijing, China) RNAiso Plus kit (Takara Bio Inc., Beijing, China), TB Green^®^ Premix Ex Taq^TM^ Ⅱ (Tli RNaseH Plus), and PeimerScript^TM^ reagent kit with gDNA Eraser were purchased from Takara Bio Inc.

### 4.2. Growth and Metabolic Activity

A micro broth dilution test was conducted to determine the influence of FLD on bacterial growth according to the Clinical & Laboratory Standards Institute (CLSI) [62]. A single colony was cultured in MH broth and was incubated with different concentrations of FLD for 24 h at 37 °C. A Multiskan Go Reader (Thermo Fisher Scientific, Waltham, MA, USA) was used to measure the absorbance at 600 nm following incubation. In continuation, *K. pnemoniae*’s metabolic activity was analyzed by the Alamar Blue (AB) assay according to the reported method [63] with minor modifications. In brief, cells from each tube were collected in 2 mL tubes and harvested by centrifugation at 10,000 rpm for 5 min, washed twice with sterile PBS (0.01 M), and suspended in 1 mL PBS (0.01 M). Then, 10 μL of AB dye (Invitrogen™, Thermo Fisher Scientific, USA) was added to the tubes and incubated at 37 °C for 1 h protected from light. PBS (0.01 mM) comprising AB dye alone was set as a blank. Absorbance was measured at OD_570nm_ and OD_600nm_ to calculate the percentage of the reduction of resazurin (blue, oxidized form) to resorufin (fluorescent pink, reduced form) attributable to cellular metabolic reduction using the following formula:(1)$$\mathrm{Metabolic}\;\mathrm{activity}\;(\%)=\frac{\left[\left(E_{\mathrm{oxi}(\mathrm{OD}570)}\;\times {\;T}_{\mathrm{OD}570}\right)\;-\;\left(E_{\mathrm{oxi}(\mathrm{OD}600)}\;\times {\;T}_{\mathrm{OD}600}\right)\right]}{\left[\left(E_{\mathrm{red}(\mathrm{OD}570)}\;\times {\;B}_{\mathrm{OD}570}\right)\;-\;\left(E_{\mathrm{red}(\mathrm{OD}600)}\;\times {\;B}_{\mathrm{OD}600}\right)\right]}\;\times \;100\%$$

E_oxi(OD570)_—extinction coefficient of AB in its oxidized form at 570 nm: 80,586.

E_red(OD570)_—extinction coefficient of AB in its reduced form at 570 nm: 155,677.

E_oxi(OD600)_—extinction coefficient of AB in its oxidized form at 600 nm: 117,216.

E_red(OD600)_—extinction coefficient of AB in its reduced form at 600 nm: 14,652.

B—blank; T—samples.

### 4.3. Biofilm Formation and Quantitative Crystal Violet Assay

The previous method was slightly modified to detect the formation of *K. pneumoniae’s* biofilms using CV staining [64]. In brief, *K. pneumoniae* cells were grown in MH broth, and different concentrations of FLD (32, 16, 8, 4, 2, 1, 0.5, 0.25 µg/mL) were incubated for 24 h at 37 °C. The nonadherent cells were removed by washing the biofilms 3 times with PBS (0.01 M) and fixed in methanol for 30 min at 37 °C. After fixation, the biofilms were stained by 0.1% CV for 30 min. The stained biofilms were rinsed with tap water to remove the unbound dye. The CV contained in biofilms was dissolved in 150 µL of absolute ethanol, and its absorbance at 570 nm was measured. The percentage of biofilms formation inhibition was calculated using the following equation:(2)$$\mathrm{Biofilms}\;\mathrm{inhibition}\;(\%)=\frac{{\mathrm{OD}}_{570}\;(\mathrm{control})\;-{\;\mathrm{OD}}_{570\;}\left(\mathrm{sample}\right)}{{\mathrm{OD}}_{570}\;(\mathrm{control})}\;\times \;100\%$$

### 4.4. Eradication of Mature Biofilms

The eradication activity of FLD against existing biofilms was also evaluated by CV staining method. Briefly, overnight *K. pneumoniae* cultures were adjusted to an OD_600_ of 0.01 and cultured at 37 °C for 24 h in 96-well plates. The bacterial suspension was slightly removed and washed twice with PBS (0.01 M) to remove planktonic bacteria. Different concentrations of FLD (128, 64, 32, 16, 8, 4, 2, 1, 0.5, 0.25 µg/mL) were added to the culture medium, with blank culture medium used as the control. The samples were then incubated at 37 °C for 48 h, followed by staining with 0.1% crystal violet to observe the disruption of biofilms caused by FLD.

### 4.5. Confocal Laser Scanning Microscopy Analysis

According to a previous report, *K. pneumoniae* biofilms were analyzed using confocal laser scanning microscopy (CLSM) [65]. Briefly, FLD was added to 10^6^ CFU/mL *K. pneumoniae* cells in MH broth, and the final concentrations of FLD were 4, 8, 16, 32, and 64 µg/mL. The media was discarded after being cultured for 24 h at 37 °C, and the nonadherent cells were removed by washing the cells 3 times with PBS (0.01 M). The bacteria biofilms were stained using a BacLight Live/Dead viability kit (L7012, Invitrogen™, Thermo Fisher Scientific, Eugene, OR, USA) according to the procedure. Afterward, excess staining was removed by washing twice with PBS (0.01 M), and then the biofilms were imaged by CLSM (LSM800, Zeiss, Jena, Germany).

### 4.6. Scanning Electron Microscopy (SEM) Analysis

To visualize the architecture of *K. pneumoniae* biofilms, SEM images were taken. The *K. pneumoniae* cells were incubated overnight at 37 °C, and then FLD was added and incubated at 37 °C for 1 h. The samples without FLD were used as a control. The cells were washed three times with sterile PBS (0.01 M), and the cells were fixed in 2.5% (*v*/*v*) glutaraldehyde for 2 h at room temperature. Following fixation, the samples were washed three times with sterile PBS (0.1 M) for 15 min each, fixed in a dark environment with osmic acid solution (1% osmic acid in 0.1 M PBS buffer), and washed three times again for 15 min each. Dehydration was conducted in ethanol–water mixtures with increasing ethanol concentrations (30%, 50%, 70%, 80%, 90%, 95%, and 100%) for 15 min each, and then isoamyl acetate rinsing was performed for 15 min. The samples were then dried using a critical point dryer (Quorum, K850, Quorum Technologies Ltd, Lewes, UK), attached to metallic stubs using carbon stickers and sputter-coated (HITACHI, MC1000, Hitachi, Tokyo, Japan) with gold for 30 s. They were finally observed and photographed with a scanning electron microscope (HITACHI, SU8100, Hitachi, Tokyo, Japan) at working distance at 13,543.33 µm, with an acceleration voltage of 3000 volts and emission current of 18,800 nA.

### 4.7. EPS Production

The EPS production was quantitatively estimated by ruthenium red staining [66]. *K. pneumoniae* cell suspensions (10^6^ CFU/mL) and different concentrations of FLD were cultured for 24 h at 37 °C (200 µL total). In each well, biofilm cells were washed 3 times with PBS (0.01 M), then stained with 0.02% ruthenium red by incubation at 37 °C for 60 min. Ruthenium red with PBS (0.01 M) served as a blank. The remaining stain solution was transferred to a new 96-well plate and the absorbance at 450 nm was measured. EPS inhibition percentage was calculated as the following equation:(3)$$\mathrm{EPS}\;\mathrm{inhibition}\;(\%)=\frac{\mathrm{AS}\;-\mathrm{AP}}{\mathrm{AB}\;-\mathrm{AP}}\times \;100\%$$

AB: absorbance of blank;

AS: absorbance of samples;

AP: absorbance of positive control.

### 4.8. Motility Assay

To learn more about the effect of FLD on *K. pneumoniae*, bacteria motility was assessed as described previously [67]. In brief, overnight *K. pneumoniae* culture was adjusted to an OD_600_ of 0.1. A semisolid agar media (0.5% MH agar) containing 4, 8, 16, and 32 µg/mL of FLD was used for the motility assay. Ten microliters of diluted bacterial solution was introduced drop-wise onto the medium surface and then incubated for 24 h at 37 °C. The halo area was used to consider motility.

### 4.9. Viable Bacteria in Biofilms

The number of bacteria in biofilms was enumerated by the plate count method [38]. Briefly, the circular glass slices were placed in six-well plates, overnight culture of *K. pneumoniae* was diluted (10^6^ CFU), and MH broth with different concentrations of FLD (4, 8, 16, 32, and 64 μg/mL) was added to six-well plates for a total 2 mL volume. After 48 h incubation, the bacterial solution was removed and washed with sterilized PBS (0.01 M) to remove planktonic bacteria, and biofilm cells were scraped using a cell scraper. Then, one hundred microliters of each dilution were spread with a sterile coating bar onto MH agar plates and incubated at 37 °C for 24 h. The number of bacterial colonies was counted and expressed in log_10_ CFU/mL. The experiment was performed in triplicate.

### 4.10. AI-2 Bioluminescence Assay

AI-2 activity was measured by bioluminescence assays according to a previous study [37]. Briefly, approximately 10^6^ CFU/mL of *K. pneumoniae* and different concentrations of FLD (4, 8, 16, 32, and 64 µg/mL) were added before the culture was incubated at 37 °C for 24 h, and centrifugated for 5 min at 12,000× *g*. The supernatant was filtered with a 0.22 µm filter. The bioluminescence reporter *V. harveyi* BB170 was incubated to an OD_600nm_ of 1.0 and then 2000-fold diluted with fresh medium. A 20 µL cell-free supernatant was mixed with 180 µL of the *V. harveyi* BB170 dilution in black 96-well plates and incubated for 3.5 h at 37 °C. Bioluminescence was measured using a multi-purpose microplate reader. Cell-free supernatant from overnight *V. harveyi* BB152 culture was set as the control.

### 4.11. Determination of Lipids, Polysaccharide Slime, and DNA in Biofilm

#### 4.11.1. Lipids

According to published methods, total lipids in biofilms were determined with a rapid colorimetric method using the sulpho-phospho-vanillin (SPV) reaction method [68]. Briefly, *K. pneumoniae* cells were grown in MH broth, and different concentrations of FLD (4, 8, 16, 32, 64 μg/mL) were cultured at 37 °C for 48 h. The biofilm cells were washed with PBS (0.01 M) and resuspended in distilled water. Then, 5 mL of vanillin reagent (120 mg vanillin, 20 mL distilled water, and 100 mL 85% phosphoric acid) was added to the cell suspension and incubated at 37 °C in shaking condition for 15 min, and 200 μL of the sample was taken and its OD_530 nm_ was measured.

#### 4.11.2. Polysaccharide Slimes

The total amount of polysaccharide slimes was estimated by the anthrone-H_2_SO_4_ reagent method as described in the previous literature [69]. Approximately 10^6^ CFU/mL *K. pneumoniae* cells were added to each well of a 6-well plate and incubated with different concentrations of FLD (4, 8, 16, 32, 64 µg/mL) at 37 °C for 48 h. After the formation of biofilms, the wells were gently washed with PBS (0.01 M) to remove planktonic bacteria. Subsequently, each well was supplemented with 1 mL of physiological saline solution after air drying. Next, the bacteria comparing the biofilms were suspended and disrupted to measure the quantity of polysaccharide slime with the anthrone-H_2_SO_4_ reagent method. Wells without FLD were set as the control group.

#### 4.11.3. Total DNA and eDNA

Based on the previously published literature [69], the effects of FLD on eDNA and total DNA in biofilms were investigated. Absorbance was measured at 260 nm following DNA extraction. Briefly, approximately 10^6^ CFU/mL of *K. pneumoniae* was added to a 6-well plate and different concentrations of FLD (4, 8, 16, 32, 64 µg/mL) were added before culturing at 37 °C for 48 h. The bacterial liquid was then removed, and planktonic bacteria were gently removed using PBS (0.01 M). The biofilms were dissolved in 1 mL of sterile physiological saline, and 200 µL of the biofilm’s cells were vortexed for 5 min to extract the biofilm matrix. After centrifugation at 12,000 rpm for 5 min, the supernatant was collected and eDNA content was measured. Total DNA was extracted from the precipitate using a DNA extraction kit, and the amount of intracellular DNA within the biofilms was measured.

### 4.12. Relative Expression of Genes

*K. pneumoniae* was incubated with FLD (32 µg/mL) for 24 h at 37 °C, and the control group was cultured in an FLD-free medium. Total bacterial RNA was extracted using the RNAiso Plus kit (Takara Bio Inc.) according to the manufacturer’s instructions. RNA quantity was measured by a NanoDrop One^C^ spectrophotometer (Thermo Scientific, USA). Further, single-stranded cDNA was synthesized using the PeimeScript^TM^ reagent kit with gDNA Eraser (Takara Bio Inc.) according to the manufacturer’s instructions. qRT-PCR was performed in a QuantStudio (TM) 6 Flex System using TB Green^®^ Premix Ex Taq^TM^ Ⅱ (Tli RNaseH Plus) (Takara Bio Inc.) according to the manufacturer’s instructions. The expression of genes was assessed by qRT-PCR with the *23S rRNA* gene as the reference gene. The primer sequences used in this experiment were synthesized by Tsingke Biotech and are shown in Table 1. Cycling parameters were as follows: pre-denaturation stage at 95 °C for 30 s; polymerase chain reaction stage at 95 °C for 5 s, 40 cycles at 60 °C for 34 s, and one melting curve stage at 95 °C for 15 s, followed by 65 °C for 1 min and 95 °C for 15 s. The 2^−ΔΔCt^ method was used to assess relative changes in gene transcription levels.

### 4.13. Safety Assessment

*G. mellonella* (purchased from Tianjin Huiyude Biotech Company) were divided into three groups at random (*n* = 8 per group) and received a single intraperitoneal injection of FLD (128, 256 mg/kg, respectively) and of PBS (0.01 M) (20 µL) at the right posterior gastropoda. Survival rates of *G. mellonella* were recorded over 120 h.

### 4.14. G. Mellonella Infection Model

The synergy between FLD and colistin was evaluated in the *G. mellonella* infection model. The larvae of *G. mellonella* were randomly divided into five groups (n = 10 per group) and four groups infected with 10 µL of *K. pneumoniae* suspension (1.0 × 10^5^ CFU/mL) at the right posterior gastropoda, while another group was treated with PBS (0.01 M). After 2 h post-infection, the *G. mellonella* were then injected with PBS (0.01 M), colistin (8 mg/kg), and FLD (16 mg/kg) or the combination of colistin (2 mg/kg) with FLD (4 mg/kg) at the left posterior gastropoda. The survival rates of *G. mellonella* were recorded for 96 h.

### 4.15. Statistical Analysis

The results of all experiments were presented as the mean ± SD of three replicates. Statistical significance was analyzed by GraphPad Prism 8.0 software (GraphPad Software, San Diego, CA, USA) with one-way ANOVA or two-way ANOVA. Significant values are represented by an asterisk: ** p* < 0.05, *** p* < 0.01, **** p* < 0.001, ***** p* < 0.0001.

## 5. Conclusions

This study verified the antibiofilm and synergistic antibacterial properties of FLD. FLD could inhibit and destroy *K. pneumoniae* biofilm formation without affecting bacterial growth. FLD inhibited the EPS production, bacterial motility, viable bacteria quantities, and QS of *K. pneumoniae*. FLD could alter the transcriptional efflux pump, LPS biosynthesis, and QS system genes that regulate biofilm formation. The synergistic antibacterial effect of FLD and colistin was also found in vivo on *G. mellonella* models. Taken together, FLD showed a promising adjuvant effect when combined with colistin for treating infections caused by *K. pneumoniae*.

## Figures and Tables

**Figure 1 ijms-25-01397-f001:**
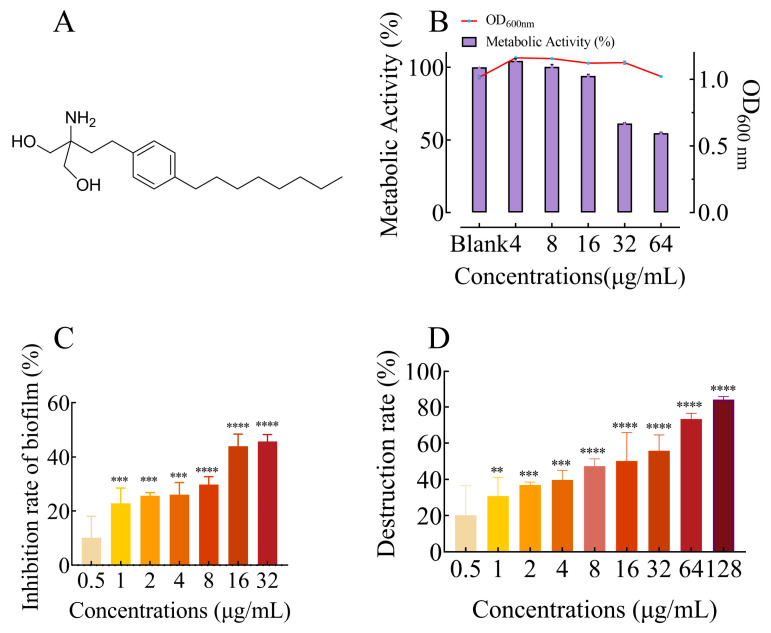
(**A**) Structure of FLD. (**B**) Influence of FLD on growth and metabolic activity of *K. pneumonia*. The line graphs and bar graph show the growth and metabolic activity of FLD, respectively. n = 3; results from all experiments are presented as the mean ± SD of three replicates. (**C**) Bar graph represents dose-dependent inhibition of *K. pneumoniae* biofilms upon treatment with FLD. Biofilm formation inhibition at various concentrations of FLD (0.25, 0.5, 1, 2, 4, 8, 16, and 32 μg/mL) for 24 h. **** = *p* < 0.0001, *** = *p* < 0.001. Results from all experiments are presented as the mean ± SD of three replicates. (**D**) Biofilm destruction at various concentrations of FLD (0.5, 1, 2, 4, 8, 16, 32, 64, and 128 μg/mL). **** = *p* < 0.0001, *** = *p* < 0.001, ** = *p* < 0.01. Results from all experiments are presented as the mean ± SD of three replicates.

**Figure 2 ijms-25-01397-f002:**
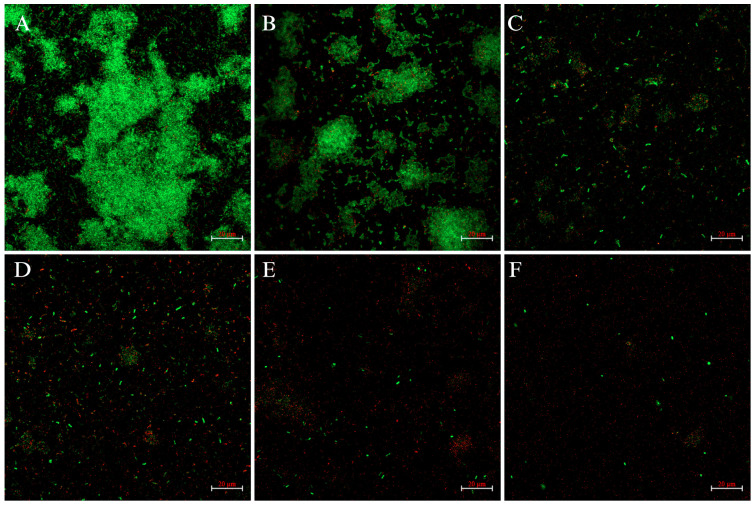
CLSM images of viable bacteria in *K. pneumoniae* biofilms ((**A**) without FLD, (**B**) treated by 4 µg/mL FLD, (**C**) treated by 8 µg/mL FLD, (**D**) treated by 16 µg/mL FLD, (**E**) treated by 32 µg/mL FLD, and (**F**) treated by 64 µg/mL FLD).

**Figure 3 ijms-25-01397-f003:**
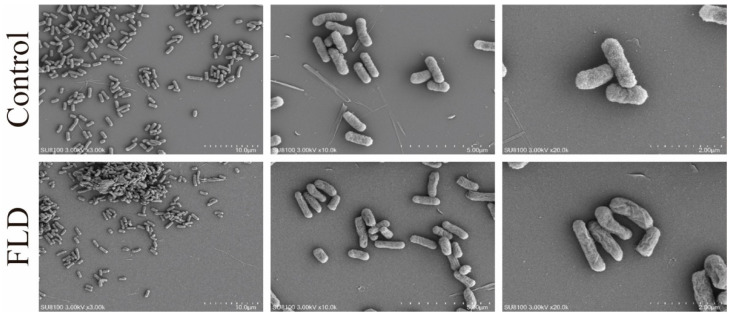
SEM images (×3000, 10,000, and 20,000) of *K. pneumoniae* morphology effected by FLD at 32 μg/mL.

**Figure 4 ijms-25-01397-f004:**
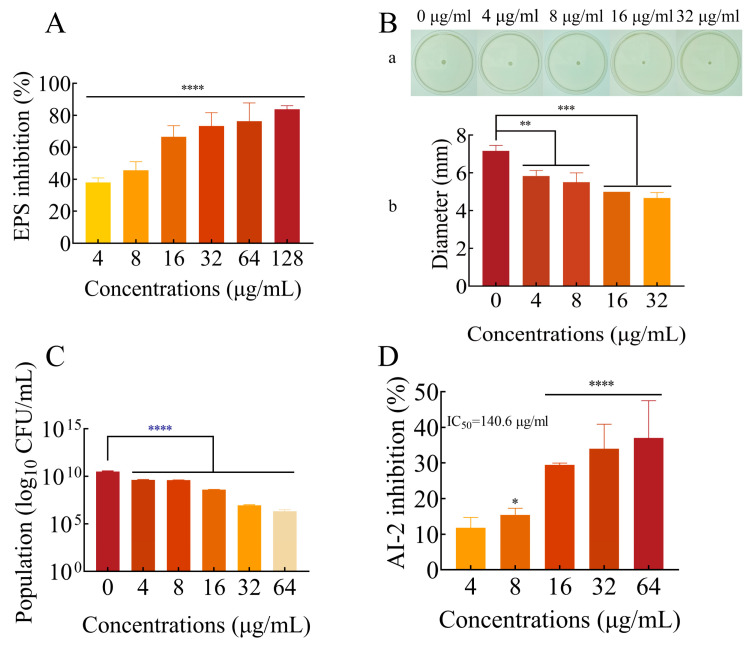
(**A**) Results of EPS inhibition (%) at various concentrations of FLD (0, 4, 8, 16, 32, 64, and 128 µg/mL) for 24 h. (**B**) Motility inhibition of *K. pneumoniae* with FLD. (a) Quantitative estimation of motility based on the diameter of the halo zone. (b) Images of motility following incubation with *K. pneumonia* at various concentrations of FLD (4, 8, 16, and 32 µg/mL). (**C**) Number of viable bacteria in biofilms under different concentrations of FLD treatment. (**D**) The inhibition of AI-2 activity was assessed by treating it with various concentrations of FLD (4, 8, 16, 32, and 64 µg/mL). n = 3. Results from all experiments are presented as the mean ± SD of three replicates. * = *p* < 0.05, ** = *p* < 0.01, *** = *p* < 0.001, **** = *p* < 0.0001.

**Figure 5 ijms-25-01397-f005:**
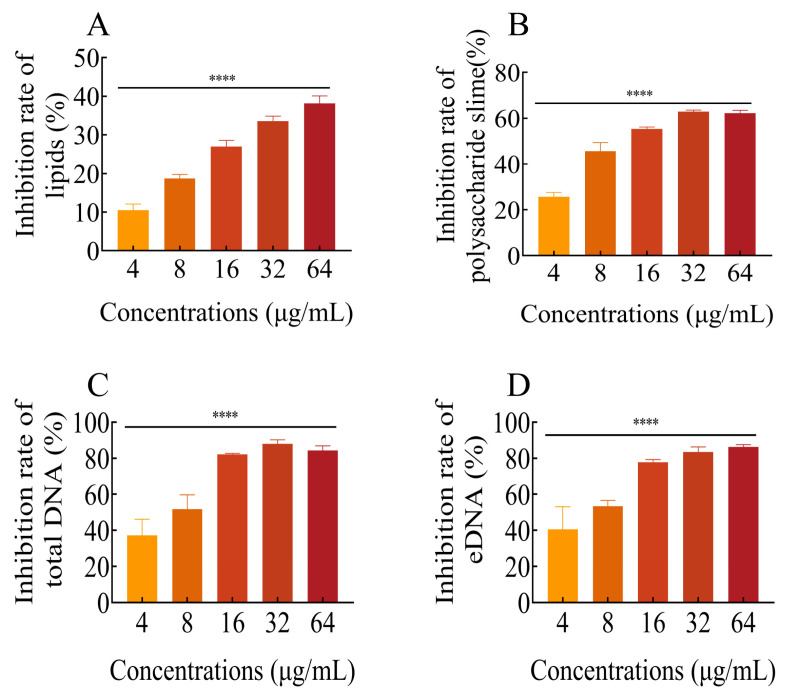
The contents of lipids (**A**), polysaccharide slime (**B**), total DNA (**C**), and eDNA (**D**) in the biofilm matrix upon FLD treatment. Each bar represents the mean ± SD of three independent experiments, **** indicates *p* < 0.0001 when compared with control group.

**Figure 6 ijms-25-01397-f006:**
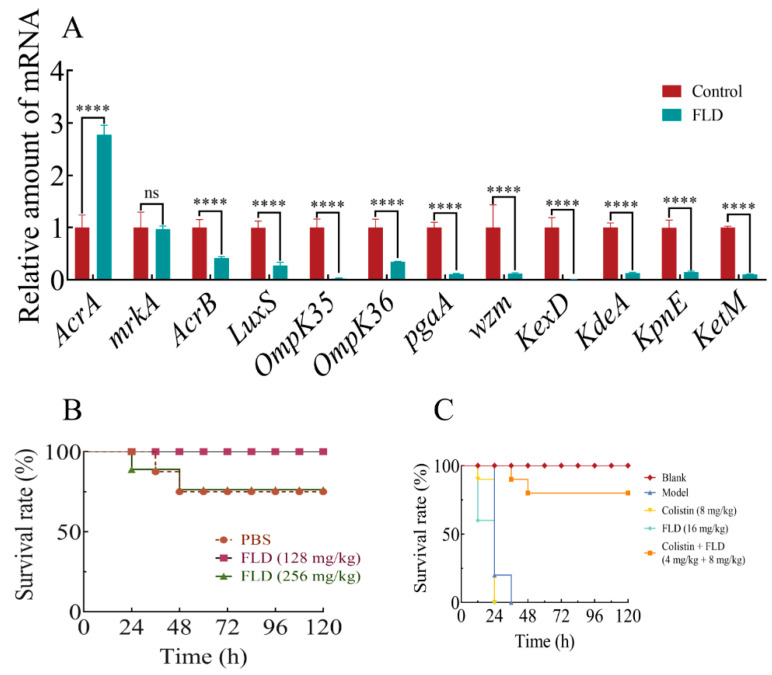
(**A**) The impact of FLD (32 µg/mL) on the transcription of biofilm-regulated genes. The qRT-PCR analysis demonstrated a noteworthy disparity in eleven genes in comparison to the control. ns indicates not significant **** indicates *p* < 0.001 when compared with control group. (**B**) The survival rate of *G. mellonella* in the three groups with different treatments for 120 h. (**C**) The survival rate of *G. mellonella*. The control group exhibited no mortality within 4 days. Conversely, the model, colistin, and FLD groups all succumbed to death within 36 h following bacterial infection. Notably, the survival rate of subjects receiving a combination treatment of FLD and colistin reached 80%.

**Table 1 ijms-25-01397-t001:** Primers used for real-time quantitative PCR.

Gene	Primer Sequence	Base Sequence	Reference
*23srRNA*	*23srRNA*-F	ATCGTACCCCAAACCGACAC	[43]
	*23srRNA*-R	TTCTCCCGAAGTTACGGCAC	
*AcrA*	*AcrA*-F	CTCTGGCGGTCGTTCTGATGC	
	*AcrA*-R	CATGTGCTGGGCTCCCTGTTG	
*mrkA*	*mrkA*-F	ACGTCTCTAACTGCCAGGC	
	*mrkA*-R	TAGCCCTGTTGTTTGCTGGT	
*AcrB*	*AcrB*-F	CAATACGGAAGAGTTTGGCA	[44]
	*AcrB*-R	CAGACGAACCTGGGAACC	
*LuxS*	*LuxS*-F	AGTGATGCCGGAACGCGG	
	*LuxS*-R	CGGTGTACCAATCAGGCTC	
*OmpK35*	*OmpK35*-F	GCAATATTCTGGCAGTGGTGATC	
	*OmpK35*-R	ACCATTTTTCCATAGAAGTCCAGT	
*OmpK36*	*OmpK36*-F	TTAAAGTACTGTCCCTCCTGG	
	*OmpK36*-R	TCAGAGAAGTAGTGCAGACCGTCA	
*pgaA*	*pgaA*-F	GCAGACGCTCTCCTATGTC-	
	*pgaA*-R	GCCGAGAGCAGGGGAATC	
*wzm*	*wzm*-F	TGCCAGTTCGGCCACTAAC	
	*wzm*-R	GACAACAATAACCGGGATGG	
*KexD*	*KexD*-F	ACCGGTTGCGCAATACCCTGA	[70]
	*KexD*-R	CGTAATTGACGCCATCCCTG	
*KdeA*	*KdeA*-F	GTTGTTCCCGTTATGTCTGGTGC	
	*KdeA*-R	CCAGCAGCCACTGTAAAAACATGC	
*KpnE*	*KpnE*-F	ATTGCTGAAATTACCGGCAC	
	*KpnE*-R	AAATACCGATCCCTTCCCAC	
*KetM*	KetM-F	TTGGCAGAGAAGGCGGTTGG	
	KetM-R	CATGACCATCCCGGGCTTG	

## Data Availability

The raw data supporting the conclusion of this article will be made available by the authors without undue reservation.

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
