# Peer review of "Fingolimod Inhibits Exopolysaccharide Production and Regulates Relevant Genes to Eliminate the Biofilm of K. pneumoniae"

_ijms, 2024, doi:10.3390/ijms25031397_

Round 1

Reviewer 1 Report

Comments and Suggestions for Authors

This is an interesting study with an important significance in multidrug resistant microorganisms era. However, some issues should be improved:

1. Please add Figure presenting chemical structure of FLD

2. The title should be modified and present full name of FLD

3. Line 18 - it should be Alamar

4. Line 490 - Candida albicans is a fungus not bacterium!

5. line 497 when instead of When

6. Line 511 - please use italics for K. pneumoniae

7. Please revise font style in references section

Comments on the Quality of English Language

None

Author Response

  1. Please add Figure presenting chemical structure of FLD

We have now added the chemical structure of FLD, and refer to it in the figure 1A.

  1. The title should be modified and present full name of FLD

The full name of FLD was presented in the title. Also, the full name of FLD was presented in the text when it appears for the first time. However, existing title will not be modified, after carefully consideration.

  1. Line 18 - it should be Alamar

We have revised the mistake in the text. (Line 18)

  1. Line 490 - Candida albicans is a fungus not bacterium!

We have revised the ″bacteria″ to ″microorganisms″ in the manuscript. (Line 491)

  1. line 497 when instead of When

We have changed the mistake in the text. (Line 498)

  1. Line 511 - please use italics for K. pneumoniae

We have changed ‘K. pneumoniae’ to ‘K. pneumoniae’ (italics) throughout the manuscript. (Line 523)

  1. Please revise font style in references section

We have modified the font style to be the same as the rest of the text.

Reviewer 2 Report

Comments and Suggestions for Authors

The paper is focused on the inhibition effects of  fingolimod (FLD) towards biofilm formation from Klebsiella pneumoniae. The topic might be interesting for readers of IJMS. The presentation and discussion of the results are accurate. I recommend the publication after the following minor revisions:

-          Figs. 2,3. Please report the scale length within the images.

-          All the equations should be numbered.

-          Conclusions section should be extended.

-          The novelty of this paper should be better highlighted in the Abstract and Conclusions paragraphs.

-          Experimental details (energy of beam, working distance,..) for SEM analyses should be reported in paragraph .6.

Comments on the Quality of English Language

Minor corrections are needed. 

Author Response

  1. Figs. 2,3. Please report the scale length within the images.

Scale bars of CLSM images have been added. SEM images were generated and have a scale.

  1. All the equations should be numbered.

The number of the formula was added according to the reviewer's comments.

  1. Conclusions section should be extended.

We have revised the conclusion section as followed:

This study verified the antibiofilm and synergistic antibacterial properties of FLD. FLD could inhibit and destroy K. pneumoniae biofilm formation without affecting bacterial growth. FLD inhibited EPS production, bacterial motility, viable bacteria quantities, and QS of K. pneumoniae. FLD could alter the transcriptional efflux pump, LPS biosynthesis, and QS system genes that regulate biofilm formation. The synergistic antibacterial effect of FLD and colistin was also found in vivo on G. mellonella models. Taken together, FLD showed a promising adjuvant when combined with colistin for treating infections caused by K. pneumoniae. (Line 617)

  1. The novelty of this paper should be better highlighted in the Abstract and Conclusions paragraphs.

 To clarify the novelty of our work, we have changed the last sentence of the Abstract as follows:

These findings suggested that FLD had substantial antibiofilm properties and synergistic antibacterial potential for colistin in treating K. pneumoniae infections. (Line 29)

For Conclusion paragraphs, the last sentence was also changed as followed:

Taken together, FLD showed a promising adjuvant when combined with colistin for treating infections caused by K. pneumoniae. (Line 622)

  1. Experimental details (energy of beam, working distance,..) for SEM analyses should be reported in paragraph .6.

We revised our manuscript by providing the details of SEM as followed:

The samples were then dried using a critical point dryer (Quorum, K850), attached to metallic stubs using carbon stickers and sputter-coated (HITACHI, MC1000) with gold for 30 s. They were finally observed and photographed with a scanning electron microscope (HI-TACHI, SU8100) at working distance at 13543.33µm, with an acceleration voltage of 3000 volt and emission current of 18800 nA. (Line 188)

Reviewer 3 Report

Comments and Suggestions for Authors

Authors proposed a paper entitled “FLD inhibits EPS production and regulates relevant genes to eliminate the biofilm of K. pneumoniae” for the publication in IJMS, mdpi.

The paper has a good scientific soundness, but it deserves to be published after major revisions. 

I suggest adding elements to the already existing  abbreviation list.

The provided abstract is quite detailed and includes a thorough overview of the research conducted. However, depending on the specific requirements or guidelines of the journal or conference you are submitting to, you might consider condensing it slightly for brevity.

The introductory section provides a comprehensive overview of the context, significance, and background of the study. It effectively introduces Klebsiella pneumoniae (K. pneumoniae), its clinical impact, and the rising concern of drug-resistant strains. Additionally, it outlines the characteristics of biofilm formation and its relevance in antimicrobial resistance. The section then transitions to the specific focus on FLD (fingolimod) and its potential antibiofilm properties. About clarity, the section is generally clear and well-organized, but consider breaking down longer sentences for improved readability. Some sentences could be simplified without losing the essential information.

The transition to the discussion of FLD is smooth, but you might consider a brief summary sentence at the end of the general biofilm discussion to tie it back to the specific focus of the study.

Section 2.2. Please check if there is a typos mistake: it could be a small typographical error in the term "K. pneumoniaes." The correct term is "K. pneumoniae" (without the 's' at the end). Correct?

Figure 1. please uniform the font size of x and y ais in all the 3 diagrams.

Line 362. Are you indicating the reference bar of the images?

Iine 440. Figure 5 seems to be Figure 51. Please have a check in the caption.

I would make some modifications in the following lines:

Line 496: "FLD significantly inhibited biofilm for-mation of biofilm formation" - consider revising for clarity.

Line 503: "FLD specifically targets the bio-film formation" - There is a hyphen in "bio-film"; it should be "biofilm."

Line 512: "antibiofilm (biofilm inhibition and elimination)" - Consider rephrasing for clarity, as it might be redundant.

Line 570: "MDR strains primarily use mrkA" - Clarify the abbreviation "MDR" upon its first mention (Multi-Drug Resistant).

Line 590: "FLD can alter the transcription of key genes regulating by affecting the" - Consider rephrasing for clarity, such as "FLD can alter the transcription of key genes that regulate by affecting."

Line 598: "in vivo and in vitro experiments demonstrated the antibiofilm properties" - Consider specifying the specific results or aspects of the experiments that demonstrated antibiofilm properties.

Line 608: "FLD alters transcriptional key genes regulating biofilm formation by affecting the efflux pump, LPS biosynthesis, and QS system" - Clarify the specific genes affected or provide a concise summary for readability.

Line 609: "These findings suggest that FLD has substantial antibiofilm properties and a novel antibacterial synergistic potential in treating K. pneumoniae" - Consider specifying the nature of the novel antibacterial synergistic potential for clarity.

Please expand conclusions together with an insight into future perspectives.

Comments on the Quality of English Language

A quite good quality of English. Only minor revisions needed.

Author Response

  1. Section 2.2. Please check if there is a typos mistake: it could be a small typographical error in the term "K. pneumoniaes." The correct term is "K. pneumoniae" (without the 's' at the end). Correct?

We have revised the mistake "K. pneumoniaes" to "K. pneumoniae's". (Line 128)

  1. Figure 1. please uniform the font size of x and y ais in all the 3 diagrams.

We have unified the font size of x and y axis in all the 3 diagrams in Figure 1, it was as followed:

  1. Line 362. Are you indicating the reference bar of the images?

Yes. The reference bar is described by this elaboration, which also serves to distinguish between images of different scales. The bacterial structures are shown in SEM images under three sets of magnification. (Line 363)

  1. Iine 440. Figure 5 seems to be Figure 51. Please have a check in the caption.

Indeed, "Figure 51" is really "Figure 5", this was a mistake. We have revised this mistake. (Line 441)

  1. Line 496: "FLD significantly inhibited biofilm formation of biofilm formation" - consider revising for clarity.

The duplication of " biofilm formation" has been deleted. (Line 496)

  1. Line 503: "FLD specifically targets the bio-film formation" - There is a hyphen in "bio-film"; it should be "biofilm."

"bio-film"” has been revised to “"biofilm". (Line 505)

  1. Line 512: "antibiofilm (biofilm inhibition and elimination)" - Consider rephrasing for clarity, as it might be redundant.

The biofilm inhibition and biofilm elimination are the two types of antibiofilm. In order to express clearly, the text in parentheses have revised as follows:

The microstructure results in conjunction with the quantitative findings provided further evidence of the inhibition and elimination activities of FLD against biofilm. (Line 511)

  1. Line 570: "MDR strains primarily use mrkA" - Clarify the abbreviation "MDR" upon its first mention (Multi-Drug Resistant).

 "Multi-Drug Resistant" has been abbreviated as "MDR" in the Abstract section. We rewrote the full name of MDR according to the reviewer’s suggestion. (Line 571)

  1. Line 590: "FLD can alter the transcription of key genes regulating by affecting the" - Consider rephrasing for clarity, such as "FLD can alter the transcription of key genes that regulate by affecting."

We have modified the sentence according to the reviewer’s suggestion. (Line 591)

  1. Line 598: "in vivo and in vitro experiments demonstrated the antibiofilm properties" - Consider specifying the specific results or aspects of the experiments that demonstrated antibiofilm properties.

We confirmed that the antibiofilm property of FLD were indeed verified in vitro, and FLD combined with colistin has a synergistic antibacterial effect in vivo. We have thus revised the original text as follows:

Overall, the antibiofilm property of FLD alone was against K. pneumoniae remarkable in vitro and the synergistic antibacterial effect of FLD combined with colistin was found in vivo. (Line 600)

  1. Line 608: "FLD alters transcriptional key genes regulating biofilm formation by affecting the efflux pump, LPS biosynthesis, and QS system" - Clarify the specific genes affected or provide a concise summary for readability.

The specific genes affected will be supplemented, and we have revised sentence as followed:

FLD alters transcriptional key genes regulating biofilm formation by affecting the efflux pump (AcrB, kexD, ketM, kdeA, and kpnE), LPS biosynthesis (wzm), and QS system (luxS). (Line 609)

  1. Line 609: "These findings suggest that FLD has substantial antibiofilm properties and a novel antibacterial synergistic potential in treating K. pneumoniae" - Consider specifying the nature of the novel antibacterial synergistic potential for clarity.

We have revised the sentence as followed:

These findings suggested that FLD had substantial antibiofilm properties in vitro, and an antibacterial synergistic potential in treating K. pneumoniae in vivo when combined with colistin. The combination can enhance the therapeutic effect by reducing biofilm-associated drug resistance. (Line 611)

  1. Please expand conclusions together with an insight into future perspectives

We have revised the Conclusion paragraph as follows:

This study verified the antibiofilm and synergistic antibacterial properties of FLD. FLD could inhibit and destroy K. pneumoniae biofilm formation without affecting bacterial growth. FLD inhibited EPS production, bacterial motility, viable bacteria quantities, and QS of K. pneumoniae. FLD could alter the transcriptional efflux pump, LPS biosynthesis, and QS system genes that regulate biofilm formation. The synergistic antibacterial effect of FLD and colistin was also found in vivo on G. mellonella models. Taken together, FLD showed a promising adjuvant when combined with colistin for treat infections caused by K. pneumoniae. (Line 617)

Round 2

Reviewer 3 Report

Comments and Suggestions for Authors

Authors responded to my issues point by points, improving significantly the quality of this paper. Now the work deserves to be published in current form.